# Changes in the Quality of Myofibrillar Protein Gel Damaged by High Doses of Epigallocatechin-3-Gallate as Affected by the Addition of Amylopectin

**DOI:** 10.3390/foods12091790

**Published:** 2023-04-26

**Authors:** Lin Chen, Rong Yang, Xiaojing Fan, Gongchen He, Zhengshan Zhao, Fangqu Wang, Yaping Liu, Mengyuan Wang, Minyi Han, Niamat Ullah, Xianchao Feng

**Affiliations:** 1College of Food Science and Engineering, Northwest A&F University, No. 22 Xinong Road, Yangling, Xianyang 712100, China; lchen@nwafu.edu.cn (L.C.); yangr0416@hotmail.com (R.Y.); fanxj1990@hotmail.com (X.F.); gche@nwafu.edu.cn (G.H.); zzs18331560943@hotmail.com (Z.Z.); fangqu199902@163.com (F.W.); yapingliu_1@hotmail.com (Y.L.); wangmengyuan@nwafu.edu.cn (M.W.); 2Lab of Meat Processing and Quality Control of EDU, College of Food Science and Technology, Synergetic Innovation Center of Food Safety and Nutrition, Nanjing Agricultural University, Nanjing 210095, China; myhan@njau.edu.cn; 3Department of Human Nutrition, The University of Agriculture Peshawar, Peshawar 25000, Pakistan; niamatullah@aup.edu.pk

**Keywords:** amylopectin, cooking loss, epigallocatechin-3-gallate, gel strength, myofibrillar protein gel, protein oxidation

## Abstract

This work investigated the improvement of amylopectin addition on the quality of myofibrillar proteins (MP) gel damaged by high doses of epigallocatechin-3-gallate (EGCG, 80 μM/g protein). The results found that the addition of amylopectin partially alleviated the unfolding of MP induced by oxidation and EGCG, and enhanced the structural stability of MP. Amylopectin blocked the loss of the free amine group and thiol group, and increased the solubility of MP from 7.0% to 9.5%. The carbonyl analysis demonstrated that amylopectin addition did not weaken the antioxidative capacity of EGCG. It was worth noting that amylopectin significantly improved the gel properties of MP treated with a high dose of EGCG. The cooking loss was reduced from 51.2% to 35.5%, and the gel strength was reduced from 0.41 N to 0.29 N after adding high concentrations of amylopectin (A:E(8:1)). This was due to that amylopectin filled the network of MP gel after absorbing water and changed into a swelling state, and partially reduced interactions between EGCG and oxidized MP. This study indicated that amylopectin could be used to increase the polyphenol loads to provide a more lasting antioxidant effect for meat products and improve the deterioration of gel quality caused by oxidation and high doses of EGCG.

## 1. Introduction

Meat and meat products are indispensable in the human diet, and their main nutritional value is to provide high-quality protein for humans. Myofibrillar proteins (MP), the most important functional protein in muscle protein, account for 50–60% of the total muscle protein and about 12% of skeletal muscle weight [1]. MP comprise an excellent gelling agent that is mainly responsible for the formation of a three-dimensional (3D) gel network of meat products during the heating process and have a decisive effect on the quality of final products [2]. Generally, MP are vulnerable to attack by reactive oxygen species (ROS) in the process of food processing and transportation, resulting in changes in the secondary and tertiary structure [3]. Therefore, the functional properties of MP will undergo irreversible changes due to polymerization and aggregation caused by oxidation, leading to poor meat product quality [4,5,6,7]. Consequently, it is of great significance to control protein oxidation in meat products.

Among all kinds of antioxidant methods, the use of antioxidants to prolong shelf life is the most effective and extensive way. Compared with synthetic antioxidants, natural antioxidants from plants have attracted much attention due to their safety and non-toxicity [8]. Generally speaking, natural antioxidants require higher doses to achieve the same effect as synthetic antioxidants, effectively protecting food proteins from oxidative attacks [9]. However, studies have found that high doses of polyphenols can interact with the protein excessively, resulting in negative changes in the physical and chemical characteristics and conformation of the protein, ultimately jeopardizing the quality of meat products [1,5,8,10,11]. Currently, there are two main mechanisms by which polyphenols modify proteins, one of which is to covalently modify proteins via thiol-quinone and amino-quinone Michael addition reactions or induce cross-linking of disulfide bonds between proteins under oxidative conditions. The thiol-quinone and amino-quinone Michael addition reactions between MP and EGCG (1000 ppm) induced a severe cooking loss and a collapsed internal structure of MP gel [5]. Gallic acid or chlorogenic acid (150 μM/g) promoted the loss of amine and thiol groups, leading to the reduced gelling ability of MP [12,13]. The thiol-quinone addition reactions between MP and excessive tea polyphenols (20 μM/g or more) were detrimental to MP, occasioning the loss of thiol groups, and deterioration of the strength and texture of MP gel [14]. Moreover, polyphenols could modify proteins non-covalently through hydrophobic, hydrogen bonding, van der Waals, and electrostatic interactions, of which hydrophobic interaction and hydrogen bonding are the most important. Non-covalent interactions between MP and rutin (200 μM/g) induced deterioration of the strength and water-holding capacity (WHC) of MP gel [15]. When there was no oxidation treatment, hydrophobic interaction and hydrogen bonding between high doses of EGCG (0.12%) and MP caused excessive protein cross-linking and had a negative impact on gel properties [16].

According to the aforementioned, interactions between polyphenols and MP are common in meat products. In particular, high doses of polyphenols caused unanticipated worsening of MP gels, as they altered the characteristics of MP [5,8,12,13,17,18]. Therefore, when polyphenols are used to replace synthetic antioxidants, how to reduce the adverse effects of high doses of polyphenols on the quality of MP gel has become an unavoidable problem in the field of meat processing. However, related studies have been rarely reported. Studies showed that EGCG-induced protein aggregation could be inhibited by glycosylation of bovine serum albumin and casein through the Maillard reaction [10,19]. However, the glycosylation reaction rate was slow under dry heat conditions, and it was difficult to control the degree of glycosylation, resulting in excessive protein denaturation, browning and the formation of anti-nutritional compounds [20]. Therefore, when polyphenols are used to substitute for synthetic antioxidants, and new theories and methods are needed to effectively control the interaction between polyphenols and proteins so as to improve the functional characteristics of proteins.

Starch is the main component of the human diet, which plays an important role in human health. It is also one of the commonly used excipients in meat products. Amylopectin is an important part of starch. In food starch, the content of amylopectin is high, generally 65–81%, which has excellent thickening, adhesion, and water-holding properties [21,22]. Waxy starch is rich in amylopectin, easy to extract, and low cost. Compared with ordinary corn starch, it has higher dilatability, transparency, viscosity, and paste stability, so it is widely used in the food industry [23]. However, corn amylopectin was selected in this study for the following reasons: (1) Waxy starch also contains amylose, and the strong interaction between amylopectin and amylose chains will affect its binding capacity with water and thermal properties [24]. (2) Waxy corn starch paste is unstable under acidic or alkaline conditions [25], and the pH value of meat products is 5.6~6.8, which limits its application in meat products. (3) Waxy starch has different molecular weights, while amylopectin has uniform and larger molecular weights. (4) Waxy starch has a sticky taste and is not acceptable to consumers. Polyphenols have polyhydroxy and carbonyl structures, so they could interact with amylopectin molecules through hydrophobic interactions, hydrogen bonds, van der Waals forces, etc.

As a result, this study aimed to improve the gel properties of MP under high doses of EGCG and oxidation conditions using amylopectin. The gel strength and cooking loss of MP gels were analyzed to understand the improvement effect of amylopectin on the quality of MP gels treated with high doses of EGCG under oxidative stress. The physical and chemical properties, conformational changes, rheological properties, and gel microstructure of MP were determined to explain the mechanism.

## 2. Materials and Methods

### 2.1. Materials

Fresh pork logissimus thoracis within 24 h post mortem was purchased from a local supermarket (Yangling, China). Epigallocatechin-3-gallate (EGCG) and amylopectin (extracted from corn) were purchased from Sigma Chemical (St. Louis, MO, USA) and Macklin Biochemical Co., Ltd. (Shanghai, China). All chemicals used in this study were of analytical reagent grade.

### 2.2. Myofibrillar Proteins (MP) Extraction

MP was collected from fresh porcine muscle according to our previously described methods [5,26]. The longissimus muscle was shredded and rinsed using 20 mM PBS (containing 25 mM KCl, 150 mM NaCl, 4 mM EDTA•2Na, and 3 mM MgCl_2_, pH 7.0). The connective tissue of the meat was removed by single-layer gauze filtration before centrifugation (2000× *g*, 4 °C, 15 min). Afterward, the precipitates were washed twice using the above buffer. The obtained pellets were washed once with 0.1 M NaCl. In the final step, MP was suspended in 20 mM PBS (pH 6.0).

### 2.3. Treatment of MP

MP was suspended in 20 mM PBS (pH 7.4, containing 0.6 M NaCl). Mixtures of amylopectin and EGCG in different proportions (Amylopectin:EGCG = 1:1, 2:1, 4:1, 8:1) were prepared and oscillated lightly in the darkness for 12 h. After that, the amylopectin/EGCG solutions were blended with MP suspensions to form different mixtures and named A:E (1:1), A:E (2:1), A:E (4:1), and A:E (8:1), respectively. The mixtures were incubated for 12 h at 4 °C with a hydroxyl radical system (0.1 mM ascorbic acid, 20 μM FeCl_3_, and 10 mM H_2_O_2_). The final concentrations of MP and EGCG in the reaction system were 40 mg/mL and 80 μM/g protein, respectively. The MP without any treatment was named Control. The MP treated only with a hydroxyl radical system were called Oxidized. The oxidized MP treated with EGCG or amylopectin were named respectively EGCG or Amp (1), Amp (2), Amp (4), and Amp (8).

### 2.4. Oxidation Degree of MP

The oxidation degree of MP was expressed as carbonyl content, which was analyzed by colorimetric method with DNPH (2,4-dinitrophenylhydrazine) [1]. Briefly, DNPH was added to the MP samples. After the addition of 20% TCA, the precipitate was washed with the mixture of ethanol and ethylacetate (1:1, *v*/*v*) three times. The MP samples were finally dissolved in the guanidine hydrochloride solution (6 M) and reacted for 20 min. Then they were centrifuged for 10 min at 4000× *g*. The absorbance of the supernatant was recorded at 370 nm. Simultaneously, the concentration of proteins in the supernatant of the control group and the experimental group was also determined by the BCA method. The molar absorption coefficient of 22,000 M^−1^ cm^−1^ was used to calculate the carbonyl content.

### 2.5. Thiol Group Content of MP

The total thiol group content of MP was estimated using the indicator DTNB (5,5′-dithio-bis (2-nitrobenzoic acid)) as outlined by Feng et al. [5]. In short, the MP samples (2 mg/mL) were dissolved in 0.1 M phosphate buffer (pH 7.4, containing 8 M urea and 3% SDS), and reacted with DTNB for 15 min. The absorbance was recorded at 412 nm. The molar absorption coefficient of 13,600 M^−1^ cm^−1^ was applied to calculate the total thiol group content.

### 2.6. Free Amino Group Content of MP

The free amine group content of MP samples was analyzed using the modified OPA (o-phthalaldehyde) method [26]. Briefly, MP samples (250 μL, 4 mg/mL) were incubated for 2 min with OPA solution (1 mL), and then the absorbance was recorded at 340 nm. The free amino group content was calculated using a standard curve produced with glycine.

### 2.7. Solubility of MP

The MP suspensions (1 mg/mL) were centrifuged at 5000× *g* (4 °C, 15 min). The protein concentration of supernatants was determined with Quick Start Bradford Protein Assay Kit (Bio-Rad Laboratories, Hercules, CA, USA). The ratio of protein concentration in the supernatant to that of the original sample was denoted as solubility (%).

### 2.8. Surface Hydrophobicity of MP

The surface hydrophobicity was measured with a hydrophobic fluorescent probe ANS (1-anilino-8-naphthalenesulfonate) [1]. The MP solutions (0.25 mg/mL, 4 mL) were thoroughly mixed with 8 mM ANS (20 μL) and incubated for 2 min. The fluorescence intensity was determined with an LS-55 fluorescence spectrophotometer (PE, Cincinnati, OH, USA) at an excitation wavelength of 390 nm. The fluorescence intensity at the emission wavelength of 470 nm was expressed as surface hydrophobicity.

### 2.9. Tryptophan Fluorescence of MP

Intrinsic tryptophan (Trp) fluorescence was used to characterize the tertiary structure of MP [5]. The fluorescence spectra of MP (0.05 mg/mL) in the range of 300–500 nm were determined by a LS-55 fluorescence spectrophotometer (PE, Cincinnati, OH, USA) with an excitation wavelength of 280 nm.

### 2.10. Secondary Structure of MP

The change of MP secondary structure was determined by Circular dichroism (CD) spectroscopy (Applied Photophysics, Surrey, UK). MP were diluted to 0.5 mg/mL with 20 mM PBS (containing 0.6 M NaCl, pH 7.4). The molecular ellipticity was measured with a 0.1 cm quartz cell at 200–260 nm [17].

### 2.11. Fourier Transform Infrared Spectroscopy (FTIR) Analysis of MP Gels

The FTIR spectrum of the MP gels was recorded with a Vertex 70 spectrometer (Bruker, Germany) [27]. The MP gels after freeze-drying were mixed with the potassium bromide powder at 1:100 and the mixtures were ground to powder and then pressed into pieces for analysis. The spectra were scanned for 32 scans with a resolution of 4 cm^−1^, and the wavelength range was 4000–400 cm^−1^.

### 2.12. Electrophoresis of MP

Sodium dodecyl sulfate–polyacrylamide gel electrophoresis (SDS-PAGE) was used to evaluate the protein cross-linking [1]. The MP samples were blended with loading buffer with or without 10% *β*-mercaptoethanol (*β*-ME) at a ratio of 4:1. The samples were boiled for 5 min. Afterwards, 10 μL of MP samples were loaded into each well. The electrophoresis mode was 30 min at 80 V and 70 min at 120 V. 0.1% Coomassie Brilliant Blue was used for gel staining. The mixture of acetic acid and ethanol (1:1, *v*/*v*) was used to elute free dyes. Gel imaging was performed in the Bio-Rad Gel Doc XRTM system.

### 2.13. Dynamic Rheological Testing of MP

The rheological behavior of the MP was characterized using a DHR-1 rheometer (Waters, Milford, MA, USA). A parallel plate with a diameter of 40 mm was used for testing. MP samples (40 mg/mL, 1.5 mL) were placed on the center of the platform and the gap between the plate and the platform was set as 1000 μm. Thermal gelation was induced at a heating rate of 2 °C/min and a temperature range of 24–80 °C. The strain and frequency were set as 2% and 0.1 Hz, respectively. Storage modulus (G′) and tan δ (loss modulus (G″)/storage modulus) were chosen as representative parameters rheological properties of MP samples [28].

### 2.14. Preparation of MP Gel

The MP samples (5 g) were placed in 10 mL beakers (50 mm height, 25 mm inside diameter). The MP gels were prepared by heating samples in a water bath. Samples were heated from room temperature (22–25 °C) to 72 °C and stayed for 10 min. After that, the MP gels were cooled to room temperature immediately and stored at 4 °C.

### 2.15. Cooking Loss of MP Gel

The total weight of the MP samples (5 g) and the beaker was recorded as m_0_ before cooking. MP gel samples were placed overnight at 4 °C, then the liquid discharged from the gel was poured out and the total weight of the gel and beaker was recorded as m_1_. Cooking loss was obtained by the following equation [29]:$$\mathrm{Cooking}\;\mathrm{loss}\;(\%)=(m_{0}-m_{1})/5\times 100\%$$

### 2.16. Gel Strength of MP Gel

The gel strength of MP gel was determined by a texture analyzer (STABLEMICVO, Godalming, UK) [29]. The probe (P/0.5) of cylinder was used for measurement with a trigger force of 5 g and a speed of 1 mm/s. The gel strength was expressed as the initial force (N) that caused the gel to fracture. 

### 2.17. Microstructure of MP Gel

A scanning electron microscope (Nano SEM-450, FEI, Hillsboro, OR, USA) was used to observe the structural features of the MP gels [1]. MP gels (0.5 cm × 0.5 cm × 0.5 cm) were fixed for 3 days with 2.5% glutaraldehyde and then washed with 0.1 M phosphate buffer (pH 7.4) four times. After that, all gel samples were dehydrated with gradient ethanol. The microstructure of the gel samples was observed by SEM after the gel samples were dried with supercritical carbon dioxide.

### 2.18. Statistical Analysis

The results were expressed as mean ± SD and repeated at least three times. The data were processed by one-way analysis of variance (ANOVA). Statistical significance among different treatments was identified by Duncan’s test using SPSS 20.0 software. *p* < 0.05 represented the difference in statistical significance.

## 3. Results and Discussion

### 3.1. Cooking Loss and Gel Strength of the MP Gels

The cooking losses of different gels areee displayed in Figure 1A. The cooking loss of natural MP gel (Control) was 31.4 ± 1.4%. After the oxidation treatment, the cooking loss significantly increased to 40.3 ± 0.7% (*p* < 0.05). Similar findings were observed by Cao et al. (2016) and Zhou et al. (2014) [2,12]. Li et al. (2021) also reported that treatment of MP with 5–20 mM H_2_O_2_ resulted in MP degradation and ultimately caused poor WHC and gel networks [30]. In this study, the treatment of •OH influenced the cooking loss of MP gels and the formation of the gel network further affected the gel textural properties. The cooking loss of the gel under oxidative stress was further increased to 51.2 ± 3.5% by the addition of 80 μM/g EGCG. This result suggested that high doses of EGCG addition increased the cooking loss of oxidized MP gels, which is consistent with the findings of Cao and Xiong (2015) [17]. This result might be due to the modification of amine and thiol groups by high doses of EGCG under oxidative stress via amine-quinone and thiol-quinone Michael addition reactions, which jeopardized the covalent cross-linking between MP and caused the gel to shrink, resulting in a worse 3D gel network. Amylopectin addition lowered the cooking loss of MP gel induced by oxidization and high doses of EGCG (Figure 1A). Amylopectin might prevent excessive aggregation between MP induced by high doses of EGCG. In addition, it could be seen from Figure 1A that amylopectin itself dose-dependently decreased the cooking loss of gels under the oxidation condition. This could be ascribed to amylopectin swelling and filling the network with MP gel after absorbing water. These results are consistent with the gel pictures (Figure 1C).

Changes in the strength of MP gels are represented in Figure 1B. Compared with the control group, the oxidation treatment had no significant influence on the gel strength (*p* > 0.05), whereas the addition of EGCG resulted in a significant increase in gel strength (*p* < 0.05). These might be owing to the gel shrinkage caused by strong hydrophobic forces and excessive cross-linking induced by 80 μM/g EGCG and oxidization. In the presence of high doses of EGCG, the addition of amylopectin dose-dependently decreased the gel strength of the oxidized MP gel, demonstrating that the structure of MP gels was improved by amylopectin. These data are consistent with the cooking loss of MP gels (Figure 1A). In the absence of high doses of EGCG, amylopectin reduced the gel strength enhanced by oxidation to some extent (Figure 1B). This indicated that amylopectin could also improve the gel properties damaged by oxidation.

From the results of cooking loss and gel strength, we suggested that amylopectin might improve the gel properties of MP by preventing protein-EGCG interactions and filling the MP gel after absorbing water and changing it to a swelling state. Zhuang et al. (2020) studied the effect of acetylated cassava starch on the gel properties of MP and came to the same conclusion [31].

### 3.2. Chemical Properties of the MP

#### 3.2.1. Carbonyl Content

Carbonyls were used to evaluate the degree of protein oxidation. Figure 2A showed the carbonyl content of all samples. The carbonyl content of oxidized MP was significantly higher than that of the control group (*p* < 0.05). The incorporation of high doses of EGCG significantly lowered the carbonyl content of MP under oxidative stress (*p* < 0.05). This result indicated that the incorporation of EGCG inhibited the formation of protein carbonyl. The carbonyl content of oxidized MP treated with EGCG did not change significantly in the presence of amylopectin compared with MP treated with EGCG alone (*p* > 0.05). This indicated that amylopectin addition did not weaken the antioxidative capability of EGCG, which still had the ability to scavenge free radicals effectively. Moreover, the incorporation of amylopectin significantly reduced the carbonyl content of MP under oxidative stress (*p* < 0.05), which suggested that amylopectin protected MP from oxidative attack.

#### 3.2.2. Free Amino and Thiol Group

The −NH_2_ groups on the side chain of MP can be translated to carbonyl groups under oxidative stress [27]. Moreover, the reaction between carbonyls and usable −NH_2_ groups can result in a further lowering of the free amine content [32]. The thiol groups of MP are easily attacked by •OH and then transformed into intramolecular and intermolecular disulfide bonds [18]. Figure 2B,C show the changes in free amino and thiol groups of MP, respectively. Compared with non-oxidized MP, the free amino and thiol content of the MP samples were significantly lost when MP were attacked by •OH (*p* < 0.05). EGCG addition did not prevent the loss of free amino and thiol groups. As a matter of fact, EGCG had the disposition to accelerate −NH_2_ and −SH oxidation and resulted in the further loss of free amino and thiol group of MP. When attacked by •OH, EGCG was translated to electrophilic quinone derivatives, and could modify −NH_2_ and −SH to form amino-quinone and thiol-quinone adducts, thereby facilitating the loss of −NH_2_ and −SH [7]. Notably, the free amino and thiol group content of MP treated with amylopectin and EGCG were higher than those treated with EGCG alone under oxidative stress. This suggested that amylopectin partially prevented the loss of free amino and thiol groups. These results indicated that the addition of amylopectin weakened the interactions between EGCG and oxidized MP to a certain extent.

#### 3.2.3. Solubility

As shown in Figure 2D, the solubility of the oxidized group was significantly lower than that of the control group (*p* < 0.05). High doses of EGCG further lowered the solubility of the MP, suggesting that the interactions between oxidized MP and EGCG led to the further aggregation of MP. The addition of amylopectin increased the solubility of the oxidized MP in the presence of EGCG, which indicated that amylopectin played a certain role in hindering the modification of MP by EGCG [33]. Noticeably, amylopectin addition also raised the solubility of oxidized MP without EGCG treatment. The result showed that amylopectin might combine with MP and sterically impede the aggregation induced by oxidation and EGCG [1].

### 3.3. Secondary and Tertiary Structure of the MP

#### 3.3.1. Tryptophan Fluorescence

The intrinsic tryptophan (Trp) fluorescence was used for monitoring the tertiary structure of the protein. Figure 3A showed the fluorescence spectra of all samples. The control group exhibited a maximum fluorescence intensity. Proteins usually appeared with a high fluorescence intensity when Trp residues were buried inside the natural protein [17]. The fluorescence intensity of oxidized MP was significantly reduced compared with the natural MP (*p* < 0.05). This was because Trp and Tyr residues were exposed to the polar environment when MP were attacked by •OH. The study by Zhu et al. (2019) also illustrated that •OH caused a reduction in the fluorescence intensity of squid MP [34]. The change of the protein conformation caused by oxidation was thought to be the main factor leading to the loss of Tyr fluorescence [24]. The fluorescence intensity of MP treated with high doses of EGCG alone under oxidative stress was the lowest among all samples. This suggested that EGCG interacted with MP and the polarity of the micro-environment around the Trp residue was altered, resulting in a further decrease in fluorescence intensity [35]. The addition of amylopectin increased the fluorescence intensity of MP in the presence or absence of EGCG under oxidative stress, suggesting that amylopectin reduced the oxidative damage of MP and the interactions between MP and EGCG. This could be ascribed to that amylopectin combined with MP and sterically impeded the damage induced by •OH and EGCG on the structures of MP.

#### 3.3.2. Surface Hydrophobicity

The surface hydrophobicity was often used to indicate the degree of protein unfolding and has a great effect on the quality of meat products. As shown in Figure 3B, the surface hydrophobicity of MP was significantly increased when exposed to •OH (*p* < 0.05), indicating an increase in the degree of unfolding. The hydrophobic amino acids were exposed on the surface under oxidative stress, increasing the binding to the probe ANS. The incorporation of EGCG did not obstruct the change of conformation. On the contrary, EGCG further lowered the surface hydrophobicity of oxidized MP, even below the level of the control group (Figure 3B). These results were consistent with the finding of Feng et al. [5]. This phenomenon could be explained by the fact that oxidization and high doses of EGCG resulted in hydrophobic aggregation owing to the excessive unfolding of proteins. Therefore, the hydrophobic area of MP was partially shielded. In addition, the combination of hydrophilic EGCG to MP also led to a decrease in the surface hydrophobicity of MP [5,36]. With the addition of amylopectin, the aggregation of MP induced by oxidation and high doses of EGCG was partially prevented, thus slightly increasing the surface hydrophobicity. Amylopectin addition reduced the surface hydrophobicity of MP under oxidative stress. These results indicated that amylopectin addition protected MP from the modifications by oxidation and EGCG, and alleviated the damage to the structures of MP (Figure 3B).

#### 3.3.3. CD Spectrum 

The secondary structure of MP was measured using a far ultraviolet CD spectrum (190–250 nm). The two negative peaks at 208 nm and 222 nm represented *α*-helical structures, which was not surprising because of the super-coiled α-helical structure at the tail of myosin. As shown in Figure 3C, these two negative peaks of MP were significantly suppressed after oxidation treatment (*p* < 0.05), indicating the unfolding of MP, which might disrupt the hydrogen bonding of *α*-helix and promote the transformation of *α*-helix to other secondary structures. This is coincident with other research [17,37]. The amount of *α*-helical structures was further decreased with EGCG treatment under oxidative stress (Figure 3C). The *α*-helical structure is mainly maintained by the hydrogen bonds between the hydrogen of amino groups (NH–) and the oxygen of carbonyl groups (C=O) within the polypeptide chain. When EGCG was oxidized to quinone, it might disturb the hydrogen bonds of MP due to the formation of amino-quinone and thiol-quinone adducts, therefore resulting in changes in the secondary structure. The excessive unfolding of MP caused by oxidization and high dose of EGCG was unfavorable for gel properties, which was coincident with the gel properties. High concentrations of amylopectin (A:E = 4:1, 8:1) increased the *α*-helix content of oxidized MP treated with EGCG. Furthermore, the *α*-helix content of oxidized MP treated with amylopectin was higher than that of oxidized MP. These results indicated that amylopectin could partially hinder the over-unfolding of MP induced by EGCG and oxidation.

### 3.4. FTIR of MP

Due to the differences in the molecular structure of different substances, the absorption of infrared radiation energy is different, and the infrared absorption spectra are different. Therefore, FTIR can be used to analyze the possible interactions between molecules. As shown in Figure 3D, the spectra of all MP gels exhibited similar patterns, no new absorption peaks were generated and no old absorption peaks disappeared in the spectra of other samples compared with the spectra of the control group. The peak in the range of 1030–1000 cm^−1^ in curves d–k was the characteristic peak of amylopectin, and the higher the concentration of amylopectin, the greater the intensity of the peak. This indicated the successful loading of amylopectin. The wide absorption band in the range of 3700–3000 cm^−1^ was assigned to hydrogen bonding [38]. EGCG addition weakened the hydrogen bonding. With the addition of amylopectin, the strength of hydrogen bonding was further weakened and the characteristic peak in the MP composite gels were slightly shifted, suggesting that amylopectin partially interfered protein–protein and protein–EGCG interactions, especially at high concentrations of amylopectin.

### 3.5. SDS-PAGE Patterns of the MP

The cross-linking of disulfide and non-disulfide bonds between proteins is important for protein gelation. Therefore, SDS-PAGE (Figure 4) was used to examine the cross-linking of MP in the presence of EGCG and amylopectin. Compared with the control group, the intensity of myosin heavy chain (MHC) and actin bands of MP was lowered after oxidative treatment, and there were more polymers on the top of the stacking gel (Figure 4A), indicating that oxidation-induced cross-linking of MP had occurred. In the presence of *β*-ME, the missing MHC and actin bands were generally recovered (Figure 4B). These results suggested that the cross-linking between MP was mainly formed by disulfide bonds. Nevertheless, residual polymers were observed on the top of the stacking gel (Figure 4B), demonstrating the presence of non-disulfide covalent bonds induced by •OH, for example, Tyr-Tyr and carbonyl-NH_2_.

The MP sample treated with high doses of EGCG did not hinder the cross-linking caused by •OH, as a matter of fact, the addition of EGCG led to the further decrease of the MHC and actin bands (Figure 4A), which indicated that EGCG induced more cross-linking of MP under oxidative stress. The MHC and actin bands were not completely recovered after *β*-ME treatment. In addition, the remnant polymers were still seen on the top of the stacking gel (Figure 4B). These results suggested that high doses of EGCG caused a certain amount of non-disulfide covalent bonds cross-linking under oxidative stress. These results combined with the analysis of free amino and thiol groups (Figure 2B,C) suggested that the formation of amine-quinone and thiol-quinone adducts between the amino and thiol groups of protein and the quinone formed by oxidation of EGCG [35].

Amylopectin had little effect on MHC and actin bands of oxidized MP treated with high doses of EGCG. Moreover, there was also no significant change in MHC and actin bands of MP treated with amylopectin alone under oxidative stress compared to oxidized MP (Figure 4). These results suggested that the addition of amylopectin had a slight effect on blocking the polymerization of MP induced by oxidation and high doses of EGCG, and therefore preventing the formation of amine-quinone and thiol-quinone adducts and protecting MP from attacks by •OH.

### 3.6. Rheological Properties of the MP

The changes of G′ and tan δ with temperature were shown in Figure 5A,B, respectively. The original MP showed a typical G′ curve and there was a peak and a trough around 52 °C and 59 °C respectively, followed by a continuous rise towards the end (Figure 5A). In the range of 45 °C to 52 °C, the rise of G′ was primarily caused by the head-head interactions of myosin, which demonstrated that the elastic gel networks were formed. Next, the G′ appeared a trough at 59 °C, demonstrating the elasticity of gels was diminished because of tail-tail interactions of myosin. This could induce the interference of the previously formed cross-linking [7,15]. Lastly, G′ appeared an uninterrupted rise from 60 °C to 80 °C owing to the strengthening of myosin head-head interactions and tail-tail interactions, and eventually, the irreversible and permanent gel structure was formed [2,15].

The oxidation treatment had no obvious effect on the G′ of the gels compared with the control group (Figure 5A), which was in agreement with the results of gel strength (Figure 1B). The final G′ of the MP gel with EGCG addition was the highest among all samples. EGCG addition might cause a tight structure instead of a homogeneous and porous gel structure, which was in accordance with the results of previous studies [5,39]. In the presence of 80 μM/g EGCG, amylopectin addition decreased the G′ of MP gels under oxidative stress. On one hand, amylopectin might hinder the excessive interactions between oxidized MP induced by EGCG, thus leading to a reduction in G′. On the other hand, amylopectin swelled after absorbing water and filled the MP gel to improve the gel properties. In the absence of 80 μM/g EGCG, amylopectin addition significantly increased the final G′. The final G′ value of the samples was not exactly in agreement with gel strength. This could be explained by that the gel strength was the result of the combined effect of shear forces, compressive and tensile, while the G′ was only representative of the shear stress [12]. The tan δ values of all samples in Figure 5B were less than 1, indicating that all MP showed elastic behavior. In general, the better the gel matrix, the lower the final tan δ value. Amylopectin lowered the final tan δ values of MP in the presence of EGCG, suggesting amylopectin improved the gelation properties of MP treated with EGCG under oxidative stress.

### 3.7. Microstructure of the MP Gels

The SEM images of different MP gels were shown in Figure 6. The pure MP gel (Control) exhibited an intact, continuous, homogeneous, and porous network structure. After oxidation treatment, the micro-structure became rough, heterogeneous, and with larger irregular pores, indicating that the oxidation was detrimental to forming a homogeneous gel network. This might be ascribed to that the MP degradation and peptide chain breakage caused by oxidative modification [2] destroyed the cross-linking of the oxidized MP, and hence reduced the gel capability of oxidized MP (Figure 1 and Figure 5). However, 80 μM/g EGCG addition exacerbated the disruption of the micro-structure of gels, leading to irregular, discontinuous protein aggregates and lump structures rather than a homogeneous and porous 3D network structure. The micro-structure was buttressed by the cooking loss and gel strength (Figure 1A,B). In the presence of EGCG, the protein aggregates disappeared and the gel network structure was gradually restored with the increase of amylopectin concentration, especially under the condition of A:E (8:1). In addition, the gels of oxidized MP treated with amylopectin alone showed a continuous and compact structure, gradually approaching that of the control. These results suggested that amylopectin protected MP from oxidative attack and ameliorated the adverse effects of high doses of EGCG.

The results of rheological properties (Figure 5) and micro-structure (Figure 6) further demonstrated the role of amylopectin in improving gel properties. Combining the results of Figure 1, Figure 2, Figure 3 and Figure 4, we believed that amylopectin might swell by absorbing water and then filling the network structure of MP gels. Moreover, amylopectin might partially prevent the interactions between EGCG and oxidized MP by steric effect.

## 4. Conclusions

High doses of EGCG altered the physiochemical and structural characteristics of MP and then jeopardized the gel properties of MP. EGCG aggravated the unfolding and destruction of MP secondary and tertiary structures under oxidative stress conditions, resulting in the loss of sulfhydryl and amino groups and MP aggregation. The solubility was reduced. MP formed a crumpled and collapsed network structure after heat induction, and so the cooking loss and gel strength were significantly increased. The addition of amylopectin (especially A:E(8:1)) weakened the excessive unfolding of MP structure caused by oxidation and EGCG, increased the content of sulfhydryl and free amino groups, and improved the solubility. A gel network structure with reduced bulk aggregates was formed after heating. Therefore the cooking loss and gel strength were reduced significantly. The mechanisms by which amylopectin improved the gel properties of MP included: The amylopectin was expansive due to water absorption, and it can improve the gel performance after filling into the gel network; In addition, the addition of amylopectin partially weakened the interaction between EGCG and oxidized MP, thus inhibiting MP aggregation. This study would provide a potential way to enhance the load of polyphenols when polyphenols were used as natural antioxidants in meat products, therefore averting their adverse impact on the functional characteristics of MP.

## Figures and Tables

**Figure 1 foods-12-01790-f001:**
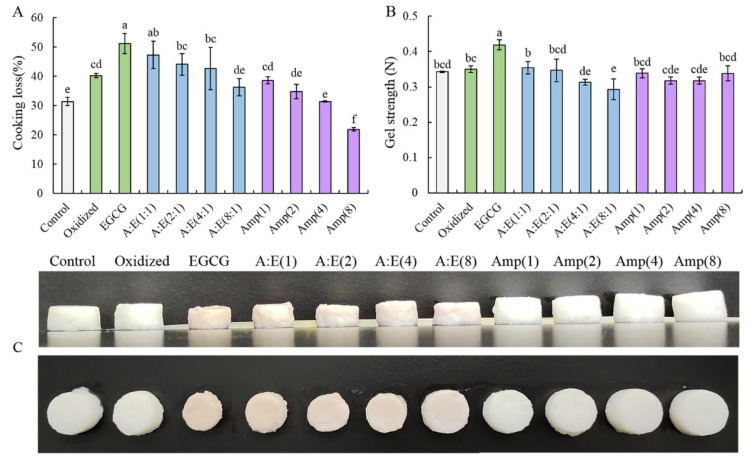
Cooking loss (**A**) and gel strength (**B**) of the MP gels as affected by EGCG and amylopectin under oxidative stress. Different lowercase letters indicate significant differences (*p* < 0.05). Visualized image of MP gel (**C**), representing Control, Oxidized, EGCG, A:E (1:1), A:E (2:1), A:E (4:1), A:E (8:1), Amp (1), Amp (2), Amp (4), and Amp (8) in order from left to right.

**Figure 2 foods-12-01790-f002:**
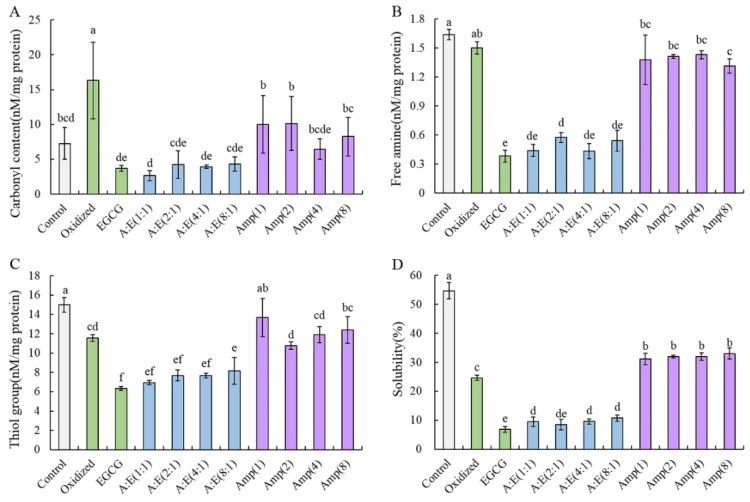
Carbonyl content (**A**), free amine (**B**), thiol group (**C**), and solubility (**D**) of MP as affected by EGCG and amylopectin under oxidative stress. Values are means ± (SD). Different lowercase letters indicate significant differences (*p* < 0.05).

**Figure 3 foods-12-01790-f003:**
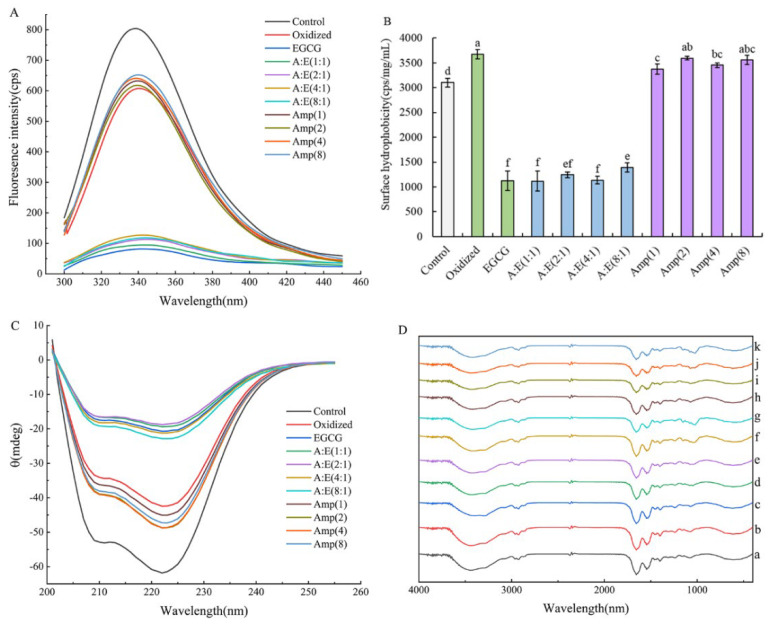
Intrinsic tryptophan fluorescence (**A**), surface hydrophobicity (**B**), CD spectrum (**C**), and FTIR spectrum (**D**) of the MP as affected by EGCG and amylopectin under oxidative stress. Different lowercase letters in (**B**) indicate significant differences (*p* < 0.05). The curves a–k in (**D**) represent Control, Oxidized, EGCG, A:E (1:1), A:E (2:1), A:E (4:1), A:E (8:1), Amp (1), Amp (2), Amp (4), and Amp (8), respectively.

**Figure 4 foods-12-01790-f004:**
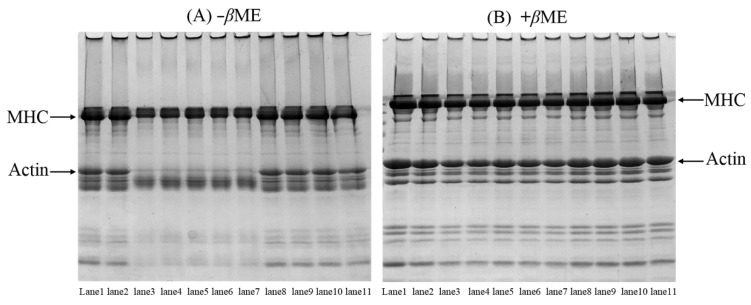
SDS–PAGE patterns of the MP as affected by EGCG and amylopectin under oxidative stress. Samples were prepared in the absence (−βΜΕ, (**A**)) or presence (+βME, (**B**)) of 10% β-mercaptoethanol. The Lane1~lane11 represent Control, Oxidized, EGCG, A:E (1:1), A:E (2:1), A:E (4:1), A:E (8:1), Amp (1), Amp (2), Amp (4) and Amp (8), respectively.

**Figure 5 foods-12-01790-f005:**
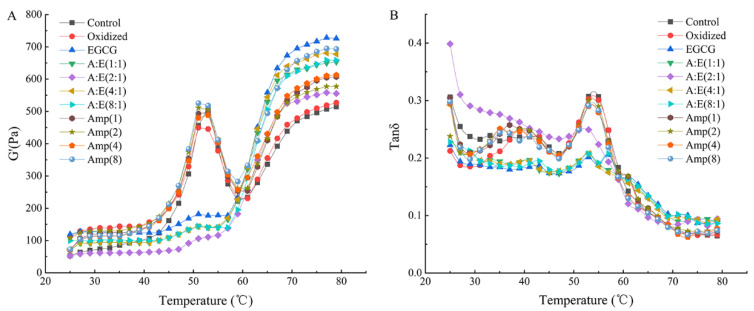
Storage modulus (G′) (**A**) and tan δ (**B**) of the MP as affected by EGCG and amylopectin under oxidative stress.

**Figure 6 foods-12-01790-f006:**
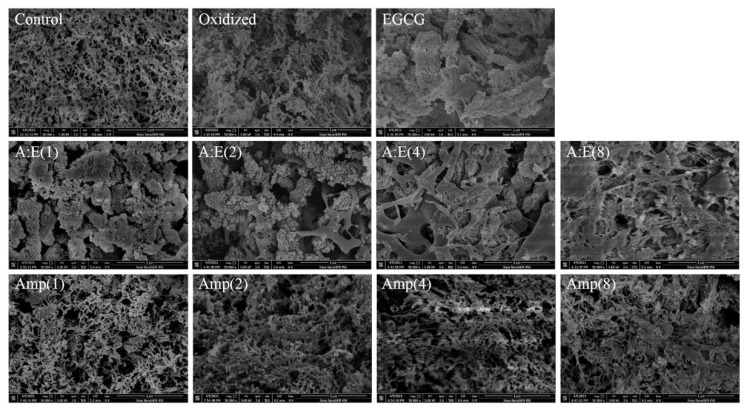
SEM images of MP gels as affected by EGCG and amylopectin under oxidative. stress.

## Data Availability

Data is contained within the article.

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
