# Peer review of "Changes in the Quality of Myofibrillar Protein Gel Damaged by High Doses of Epigallocatechin-3-Gallate as Affected by the Addition of Amylopectin"

_foods, 2023, doi:10.3390/foods12091790_

Round 1

Reviewer 1 Report

foods-2300227-peer-review-v1

Title:  Improvement of amylopectin addition on the quality of myofibrillar protein gel damaged by high doses of epigallocatechin-3-gallate

Comments to Authors: Lin Chen, Rong Yang, Xiaojing Fan, Gongchen He, Zhengshan Zhao, Fangqu Wang, Yaping Liu, Mengyuan Wang, Minyi Han, Niamat Ullah, Xianchao Feng

The Authors were  investigated the improvement of amylopectin addition on the quality of myofibrillar proteins (MP) gel damaged by high doses of epigallocatechin-3-gallate (EGCG, 80 μM/g protein). This study indicated that amylopectin could be used to increase the polyphenol loads to provide more lasting antioxidant effect for meat products

Methods

Methods presented correctly, a wide range of research.

Results

Results well described and discussed

Author Response

The Authors were  investigated the improvement of amylopectin addition on the quality of myofibrillar proteins (MP) gel damaged by high doses of epigallocatechin-3-gallate (EGCG, 80 μM/g protein). This study indicated that amylopectin could be used to increase the polyphenol loads to provide more lasting antioxidant effect for meat products

1.Methods

Methods presented correctly, a wide range of research.

Response:

Thanks for your positive comments.

2.Results

Results well described and discussed

Response:

Thanks for your positive comments.

Reviewer 2 Report

Article “Improvement of amylopectin addition on the quality of myofibrillar protein gel damaged by high doses of epigallocatechin-3-gallate” highlights the applications of amylopectin in the stabilization of protein structure. I think that authors should try to include a scientific discussion and write clearer conclusions. Different points must be considered.

1. Add the numerical data in the abstract portion.

2. Mention the full name of EGCG in the keywords.

3. In introduction section justify the objective of this study, amylopectin isolation a costly process, if author use waxy starch in palace of amylopectin then this process will be cost effective, justify.

4. Authors write the manuscript like a report, add scientific discussion for data analyzed. Add possible reasons for variation observed in the different samples. Also compare the results with previous studies.

5. Authors showed the results in Figure form; if possible add some tabular data.

6. If possible add the textural properties of samples, as these are more important than protein stabilization.

7. Conclusion is just the summary of results. Please talk about the big picture and the major findings of the work. What's new that this paper offers to the readers?

Author Response

Article “Improvement of amylopectin addition on the quality of myofibrillar protein gel damaged by high doses of epigallocatechin-3-gallate” highlights the applications of amylopectin in the stabilization of protein structure. I think that authors should try to include a scientific discussion and write clearer conclusions. Different points must be considered.

Response: We appreciate the reviewer’s kind suggestions. We have revised the discussion after consulting relevant literatures and giving a deep probing into the results. We are sorry that we did not write clear conclusion. We have revised the conclusion to be clearer through discussing the results in detail. We have revised the introduction, and provided sufficient background and added relevant references. We hope the introduction is complete now.

1.Add the numerical data in the abstract portion.

Response: We are very sorry that we did not take into account that the summary section should present numerical data. It is important to present numerical data in abstract portion to make the abstract clearer and more scientific. So, we have revised the content in the abstract portion and added the numerical data.

  1. Mention the full name of EGCG in the keywords.

Response: We appreciate the reviewer’s kind suggestion. The full name of EGCG is Epigallocatechin-3-gallate, and we have revised EGCG to its full name in the keywords section.

  1. In introduction section justify the objective of this study, amylopectin isolation a costly process, if author use waxy starch in palace of amylopectin then this process will be cost effective, justify.

Response: We are sorry that we didn't consider the cost carefully. Waxy starch is indeed more cost effective than amylopectin and contains a higher content of amylopectin. But amylopectin also has its advantages and significance. We have justified the objective of this study in the introduction section. In addition, Our study showed that amylopectin can inhibit protein-polyphenol interaction, thereby improving gel properties damaged by high doses of EGCG. Therefore, amylopectin is an effective additive. Compared with pure amylopectin, Waxy starch also contains amylose. The effects of amylose on the gel properties of MP is not clear. So, Amylopectin is more effective in improving gel properties and requires less to achieve the same effect, thus reducing the relative cost. We can also reduce cost by extracting amylopectin by simple and convenient ways, such as warm water extraction method.

  1. Authors write the manuscript like a report, add scientific discussion for data analyzed. Add possible reasons for variation observed in the different samples. Also compare the results with previous studies.

Response: We appreciate the reviewer’s kind suggestions. We are sorry that we did not give the scientific discussion for data analyzed. We have revised the discussion after consulting relevant literatures and giving a deep probing into the results. Also, we have added possible reasons for variation observed in the different samples. We hope that the discussion is more scientific now.

  1. Authors showed the results in Figure form; if possible add some tabular data.

Response: We think that the figures are more vivid and explicit than tables, so we showed the results in figure form. And we will change the figures into tables if the reviewer insist that we should add some tabular data.

  1. If possible add the textural properties of samples, as these are more important than protein stabilization.

Response: We appreciate the reviewer’s kind suggestions. But we want to specifically study the change of textural properties of meat products in the later study. There are mainly two main reasons. On one hand, this paper mainly focuses on the study of mechanism. On the other hand, the study on the quality characteristics of meat is a relatively systematic study, including textural properties, water retention, chroma, digestibility, micro-structure, etc.

  1. Conclusion is just the summary of results. Please talk about the big picture and the major findings of the work. What's new that this paper offers to the readers?

Response: We appreciate the reviewer’s kind suggestions. We are sorry that we did not talk about the major findings and new points of the work in conclusion portion. We have revised the content in conclusion portion after thinking about the major findings and new points of the work. We hope that the conclusion be complete and reasonable now.

Reviewer 3 Report

The manuscript provides interesting data about the changes in the qualities of myofibrillar proteins as affected by the addition of amylopectin. In this regard, my opinion is that the title of the paper is not well formulated, because the authors study the effect of the addition of amylopection on the qualities of myofibrillar proteins but they do not improve the amylopectin itself. So I suggest that the title be changed as Changes in the quality of myofibrilar protein gel damaged by high doses of epigallocatechin-3-gallate as affected by addition of amylopectin.

The section materials and methods might be improved. For instance in line 100 "fresh back longissimus of pig carcasses" should be corrected to "fresh pork logissimus thoracis et lumborum muscle".

The authors should extend the description of the carbonyl assay and add the reference. I my opinion they have missed an essential step of the method that is read the absorbance of the protein sample in the ultraviolet spectre of determine the quantity of the protein need to calculate the carbonyl content.

It will be good to add references for all the methods applied in the study.

Author Response

The manuscript provides interesting data about the changes in the qualities of myofibrillar proteins as affected by the addition of amylopectin. In this regard, my opinion is that the title of the paper is not well formulated, because the authors study the effect of the addition of amylopection on the qualities of myofibrillar proteins but they do not improve the amylopectin itself. So I suggest that the title be changed as Changes in the quality of myofibrilar protein gel damaged by high doses of epigallocatechin-3-gallate as affected by addition of amylopectin.

Response: We appreciate the reviewer’s kind suggestions. We think it is a good idea to change the title as “Changes in the quality of myofibrilar protein gel damaged by high doses of epigallocatechin-3-gallate as affected by addition of amylopectin”. This new title is indeed more reasonable.

  1. The section materials and methods might be improved. For instance in line 100 "fresh back longissimus of pig carcasses" should be corrected to "fresh pork logissimus thoracis et lumborum muscle".

Response: We are sorry that we did not correctly write the name of material. We have checked full manuscript and revised the description with errors. We hope that the names and descriptions are reasonable and correct now.

  1. The authors should extend the description of the carbonyl assay and add the reference. I my opinion they have missed an essential step of the method that is read the absorbance of the protein sample in the ultraviolet spectre of determine the quantity of the protein need to calculate the carbonyl content.

Response: We appreciate the reviewer’s kind suggestions. We are sorry that we did not amply describe the carbonyl assay and missed an essential step of the method. We have complemented the method completely, and we have checked the full method section and made a supplement. We hope that the method section is more complete now.

  1. It will be good to add references for all the methods applied in the study.

Response: We appreciate the reviewer’s kind suggestions. It is very important to cite references in the method section to justify the method. So, we have revised the method section, and added corresponding references. We hope that the method section is better now.

Round 2

Reviewer 2 Report

Authors address all the comments suggested by me. This article may be accepted in this esteemed journal.